# Usability and psychometric properties of a battery of tools to assess intelligence, executive functioning, and sustained attention in Tanzanian children

Georg Loss[1,2], Hannah Cummins[3], Nicolaus Gutapaka[4], Jane Nyandele[4], Sylvia Jebiwott[3], Deborah Sumari[4], Thabit Athuman[4], Omary Juma[4], Susanne P. Martin-Herz[5], Ally Olotu[4], Michelle S. Hsiang[3,5,6]*, Günther Fink[1,2]

1 Swiss Tropical and Public Health Institute, Switzerland, 2 University of Basel, Switzerland, 3 Malaria Elimination Initiative, Institute of Global Health Sciences, UCSF, U.S.A, 4 Biomedical Research and Clinical Trials Department, Ifakara Health Institute, Tanzania, 5 Department of Pediatrics, UCSF, U.S.A, 6 Department of Epidemiology and Biostatistics, UCSF, U.S.A

* Michelle.Hsiang@ucsf.edu

**Data Availability Statement:** All relevant data are within the paper and its Supporting Information files.

## Abstract

### Background

Measuring neurocognitive functioning in children requires validated, age-appropriate instruments that are adapted to the local cultural and linguistic context. We sought to evaluate the usability and psychometric properties of five tools that assess general intelligence, executive functioning, and sustained attention among Tanzanian children.

### Methods

We adapted five age-appropriate neurocognitive assessment batteries from previously published assessment materials to the Tanzanian context. We enrolled children 6 months to 12 years of age residing in the rural ward of Yombo, Pwani Region. Feasibility and acceptability of all instruments was assessed qualitatively and quantitatively, including measurement of refusal rates, ceiling or floor effects, and time requirements. We assessed internal consistency using Cronbach's alpha and convergent validity using standard correlation analysis. Score gradients across age were explored using polynomial regression analysis.

### Findings

All five instruments required minimal adaptations to the Tanzanian context. Two-hundred sixty one children aged 6 months to 12 years completed the assessment. Refusal rates were consistently low (5.9% at the highest) and no ceiling or floor effects of measurements were observed. Feedback from assessors and caregivers indicated adequate test durations and generally high acceptability of instruments. All instruments showed good internal consistency with Cronbach alphas at least 0.84 for all tests. We found satisfactory convergent validity; all test scores strongly correlated with age.

**Funding:** This work was supported by NIH/NIAID grant # 1U01AI155315-01A1.

**Competing interests:** NO authors have competing interests.

## Conclusion

The five instruments identified to assess general intelligence, executive functioning, and sustained attention constructs in Tanzanian children seem to work well in this setting.

## Introduction

Assessing neurocognitive functioning in children presents a complex challenge, particularly in contexts with limited resources [1–6]. Instruments need to assess a broad spectrum of cognitive constructs, ranging from basic perceptual and attentional abilities to higher-order executive functions such as problem-solving and decision-making, while simultaneously ensuring age-appropriateness and cultural relevance. There is ample evidence that instruments measuring cognitive constructs are not applicable transculturally (i.e., across different economic, cultural, and linguistic groups) unless specifically designed in a transcultural way [1–4, 7–10]. Further, psychometric evaluations must not only be validated for specific age groups and settings, but also must be able to discriminate between individual ability levels with sufficient granularity to detect meaningful differences in clinical and programmatic evaluations.

The motivation of the herein presented work was to adapt and validate instruments measuring neurocognitive development among children 6 months to 12 years of age participating in the CHILD (Child Health and Infection with Low Density Malaria, clinicaltrials.gov: NCT05567016) [11] intervention study. CHILD aims to investigate the clinical and economic value of more sensitively identifying and treating low-density malaria infections in Tanzanian children, including the impact on their neurocognition. Low-density malaria infections (LMI) refer to cases where the level of parasitemia is detectable by highly sensitive molecular methods such as PCR but undetectable by standard diagnostic tests. These infections were previously considered asymptomatic and beneficial for inducing immuno-protection against clinical malaria [12–15]. But more evidence, particularly from low transmission settings, showed negative health effects and unclear immuno-protective effects [16–21]. While severe malaria infections have long been recognized for their potential neurological complications [22], the neurological sequelae of LMI are less understood. Chronic exposure to the malaria parasite, even at low levels, may negatively impact cognition, specifically the functions of attention and abstract reasoning, in addition to more generally worsening school outcomes [23–27]. Consequently, LMI may have significant long-term implications for academic performance, educational attainment, and overall quality of life.

This work aims to provide insights into measuring neurocognitive functioning and attention of children in Tanzania, ultimately contributing to the evaluation of the CHILD intervention study. We provide a detailed description of the tools and adaptations, assess their local feasibility and acceptability, and rigorously evaluate their reliability and validity in capturing the complexity of neurocognitive processes.

## Methods

This is a cross-sectional validation study conducted in Yombo, Tanzania in 2023. We selected the Yombo ward for its proximity and similarities to the CHILD study area, which is conducted in the adjacent Fukayosi and Kiwangwa wards of the same low malaria transmission region. Yombo shares several relevant characteristics with Fukayosi and Kiwangwa, including demographic makeup, rural settings, socio-economic conditions, and malaria transmission

rates. Yombo is rural, consisting of 31 Hamlets with an estimated total population of 11,102 based on the most recent 2022 census.

## Sampling

We aimed to enroll a locally representative sample of 20 children per age year group between infancy (≥6 months) and 12 years inclusive, for a total sample of 260 children. Following the eligibility criteria of the corresponding CHILD study, children with known history of chronic illness requiring specialty care including diabetes mellitus, cancer, or stage 3 or 4 HIV/AIDS or not consenting were excluded. Six hamlets (Chamkwela, Chombe, Kikuichora, Lwazi, Matimbwa, Ushokolani) were randomly selected for a complete household listing. This recruitment listing took place between 31 March and 7 April 2023. All households in the hamlets were approached, and the name of the household head and names and ages of children below 13 years were recorded from an adult household respondent. Four hundred eighty-eight children in 252 households were listed in total. Twenty-five children per age group were randomly selected for inclusion with the exception of age 0, where all 11 available children selected in the group of <1 year olds (see S1 Table in S1 Appendix). The children's ages recorded during the household listing compared to the subsequent study assessments differed slightly, as the ages of children in the household were reported less accurately on average during the household listing. We did not conduct a formal sample size calculation, as the primary objective of this pilot study was to assess the feasibility and acceptability of multiple measurement tools in the local intervention study context.

## Instrument selection

To develop fit-for-purpose neurocognitive instruments in the framework of the CHILD study, we selected existing Kiswahili-translated instruments previously adapted for use in Tanzania to cover our study's wide age range. The instruments needed to be feasible for implementation by health workers with at least an undergraduate education. Other considerations for instrument selection were: history of use in low-income and rural setting, inclusion of constructs relating to CHILD study objectives, cultural and age-appropriateness, previous validation in the local context, simplicity and ease of use (curation, administration, and analysis), and cost efficiency. Based on these criteria, three instruments assessing cognitive function and two assessing sustained attention were selected. We then adapted and evaluated the psychometric properties of the selected instruments as follows.

## Global Scales for Early Development: Long Form

The Global Scales for Early Development (GSED) [9] is a standardized assessment tool designed to evaluate the cognitive, language, motor, and social-emotional development of children aged 0 to 42 months of age in low- and middle-income countries. The GSED Long Form is a directly administered assessment recording assessor observed child interactions and behaviors. It contains a total of 155 items with age-specific starting points, and performance specific forward, skip and backward rules. On average, around 30 items need to be administered on average to complete the assessment. An overall development score (D-score) aggregating information on skills and developmental milestones is the derived from child's responses to items and the difficulty of those items. GSED is culturally adaptable and available in multiple languages. A validated Kiswahili version of the GSED assessment battery materials (release version 1.0) as well as GSED collection tools (pre-release directly from the GSED developers) were utilized for this study [28, 29].

## International Development and Early Learning Assessment

The International Development and Early Learning Assessment (IDELA) [8] consists of 24 primary items that directly assess developmental skills and early learning in pre-school children aged 3 to 6 years. IDELA has been validated in multiple low- and middle-income countries, and Kiswahili versions of all IDELA materials were available directly from the creators' repository [30]. This instrument evaluates four developmental subdomains (motor development, emergent numeracy, emergent literacy, and social-emotional development) along with two additional developmental areas of executive functioning and self-regulation, which are brief and were excluded from our main analysis. The IDELA total score is expressed as the average of proportions achieved in the four domains as assessed using 22 items.

## The East African Cognitive Assessment Battery

The East Africa Cognitive Assessment Battery (EACAB) [7, 31] is a previously validated tool for evaluating core constructs of cognitive ability and behavioral problems in school aged children in low-resource settings of East Africa. The battery consists of Kiswahili-translated adaptations of established instruments (e.g., Kaufman Assessment Battery for Children 2$^{nd}$ Ed, Koh's Block, Brief-P: Behavior Rating Inventory of Executive Function Preschool, SDQ: Strengths and Difficulties Questionnaire). For our study, we incorporated sub-tests from the EACAB covering critical general intelligence constructs of: i) learning, ii) simultaneous processing, iii): sequential processing, iv) planning, v) verbal intelligence, and iv) selective and sustained attention, as well as the executive functioning constructs of: i) working memory EFWM1 (Osterrieth Complex Figure Test) and ii) working memory EFWM2 (Osterrieth Complex Figure Recall Test). Scores for these tests are the number of items correctly completed, with some tests having a time limit. Tests with increasing difficulty employ stopping rules activated when minimal scores at defined sections in a test are not achieved, thus limiting the assessment duration and burden.

## Tests for executive function: Sustained attention

We assessed sustained attention using two different locally adapted tests. First, we used the Sustained Attention Pencil Tap Test (SAPT), which has been used in children 5–10 years of age [32]. During SAPT, the child is instructed to copy the assessor's tapping of a pencil against a hard surface, e.g., the table, while coloring shapes using her pencil. The assessor records the tapping responses and intermediate random taps to assess the child's ability to maintain focus, sustain effort, and resist distractions over the duration of the task. Second, we used the Sustained Attention Code Transmission Test (SACT), a sustained attention sub-test of the Test of Everyday Attention for Children battery (TEA-Ch) [26, 33, 34] designed for children aged 8–15 years. The test follows an n-back process, where the tested child listens to a 10 minute monotone pre-recorded sequence of (seemingly random) digits read aloud at the speed of one per second. The child is required to listen for the occurrence of a specific sequence (two consecutive occurrences of the number 5) and to speak the number which immediately preceded the code. Similar to the pencil tap test, the number of correct responses allows for assessing the child's ability to maintain focus. For both tests, the final score is derived as the sum of correct items, while the SAPT is additionally adjusted for the number of intermediate random taps.

## Pre-testing of instruments

All instruments and corresponding data collection tools were pretested by trained local assessors with 30 children recruited between 8 February and 27 March 2023. In accordance with

ethical guidance and approvals, the assessors obtained verbal informed consent from each parent or guardian of participants. Kiswahili-translated and Tanzania-adapted versions were readily available for all instruments. Pre-testing focused on assessing local cultural appropriateness, acceptability and feasibility of administration, as well as minimization of assessment duration to no more than 60 minutes. We further aimed to identify potential threats or challenges for implementation. For each test, these themes were assessed by administering standardized questionnaires among assessors, participants, and accompanying caregivers. GSED was pre-tested in six children 6 months-3 years of age and IDELA was pre-tested in six children 3–5 years of age. The selected EACAB sub-tests were pre-tested in 18 children 6–12 years of age. The Pencil Tap Test and the Code Transmission Test were pretested in 12 children 4–8 years of age and 10 children 8–12 years of age, respectively.

## Pre-test results and adaptations of tools to local context

The acceptability of GSED and IDELA during qualitative assessments was high; minimal changes to the instruments were required. Based on feedback from the assessors and several pre-test participants' parents, IDELA tool items concerning residential location and empathy were adapted in order to accommodate locally accepted responses in Kiswahili. Previously, the GSED was validated for children aged up to 36 months [35] and the IDELA for children aged 42 months to 71 months inclusive [8]. Here, the GSED and IDELA were pre-tested in children aged 6 to 47 months inclusive (i.e. 0.5 to <4 years), and children aged 36 to 71 months inclusive (i.e. 3 to <6 years), respectively, with overlap to optimize age cut-offs for the CHILD study context.

Implementation of the pre-selected seven EACAB sub-tests was time consuming, especially for older children for whom the duration of testing often exceeded 90 minutes. To reduce test duration, the test for selective/sustained attention, which was covered by our two ad-hoc tests (described below), as well as the verbal intelligence test, which demonstrated the lowest participant acceptability, were dropped.

For the two SAPT and SACT tests, we made several changes based on qualitative pre-testing results. Instructions were expanded and improved to promote participant understanding. The Code Transmission Test originally used entirely pre-recorded instructions such that practice and test activities were played out loud for the test child. These recordings were not well-understood by children during pre-testing due to the rapidity of playback speed and the absence of pauses for questions or clarifications. Given this, the instructions were instead transcribed to be read aloud by the assessor, and separate audio files were generated of the test activity (i.e. pre-recorded enumeration of the test number sequences) for the assessor to play in segments. This would ensure consistency of the core test while enhancing flexibility and responsiveness to individual participants during the instructional phase. Practice scripts were adapted from two Kiswahili versions of the Code Transmission Test [26, 34]. For the SAPT, a stop rule was added, modeled after the SACT (i.e., the test stops after five missed test items). For both sustained attention tests, the scoring sheets were adapted to correspond to the digital collection tool, where only the final scores were collected (while item scoring sheets were only scanned for documentation). Finally, the SAPT was employed for children aged 5–10 years and the SACT for children aged 7–12 years with the overlapping ages of 7 to 10 years intended for age-calibration. The sequences of tests in the final age-related test batteries were equal for all participants.

## Staff and training

We recruited assessors from medical personnel (physicians and nurses) associated with the CHILD intervention study. Extensive training materials were available from the respective

instrument creator's repositories. Eleven assessors were trained in full between 29 March and 6 April 2023, shortly before the start of the validation study. The training progressed from theoretical background on the instruments to applied practice in administering the tools to test children both in the training facility and in the community, recreating the planned validation study setting. Training sessions were optimized to minimize redundancy, such that teaching materials with shared characteristics were paired or combined. The training further included hands-on use of tablet-based data collection applications and standardized instruments, which simplified test administration and data collection.

## Data collection

The validation study took place between 24 April and 11 May 2023. During a single home visit, assessors enrolled participants based on the household listing selection, and collected their information. A household was approached three times if non-responsive before being excluded. During the visit, assessors obtained written individual informed consent from a parent or guardian of the participating child in accordance with ethical guidance and approvals. We stopped enrollment when our goal of 260 children was achieved. Data was collected digitally on tablets (Android-based operating system) using the Open Data Kit (ODK) [36] Collect application and the stand-alone ODK-based GSED application provided by its developers. These applications substantially reduced the burden on assessors, as test logic and scoring were pre-programmed and appropriate tests, items and related instructions were automatically selected and displayed. The assessors recorded children's engagement in neurocognitive assessments and documented any questions or concerns expressed by the children or caregivers. The study coordinator regularly monitored all study personnel in the field to guarantee ongoing adherence to study protocols.

## Variables

We collected the following child and socio-demographic information: child's gender (male vs. female), highest education achieved by maternal and primary caregiver (below primary, primary, secondary, or tertiary attainment), a proxy for household income [37–39] (i.e. 19 site-specific household assets—see S2 Table in S1 Appendix), household provision of child stimulations for children aged < 7 years (UNICEF's Multiple Indicator Cluster Survey (MICS)), current school attendance (yes vs. no, including kindergarten, pre-school or primary school), respondent-perceived school performance compared to other pupils at the same school (much worse, somewhat worse, same, somewhat better, much better), and child anthropometry (weight and height). Height in children under 24 months of age was measured using a Seca length board, while in those 24 months or older was measured using a portable Seca stadiometer with weighing scale. The assessors recorded child height with a precision of 0.1 cm. A Seca weighing scale balance was used to weigh the children. For children below 2 years, the weight was derived from the difference of weighing the parent or guardian respondent with and without child.

In addition, we collected quantitative information on the acceptability of the neurocognitive assessments from participants and assessors, namely perceived difficulty of assessments by test children (when able to respond) or accompanying caregiver (from 1 = very easy, 2 = easy, 3 = moderate, 4 = difficult, 5 = very difficult), assessor rated level of disturbance (1 = none to 5 = severe) as well as occurrence of pre-defined disturbances (loud environment, disturbance by other children/adults, shyness/fear, child engagement). We further collected qualitative information using open-ended questions on any other feedback or concerns regarding the assessments.

## Statistical analyses

We summarized characteristics of the study population using counts and percentages or means and standard deviations of key socio-demographic variables as appropriate.

To evaluate all instruments, we followed a multi-step process. First, we assessed the feasibility of each instrument in terms of participant refusals, defined as completion of fewer than 50% of test item questions per instrument. For tests where non-completion of items was part of the scoring (i.e. EFWM and sustained attention tests), refusal was defined as proportion not attempting the assessment from the start. Next, the instruments' floor and ceiling effects were expressed as proportions of children scoring the respective minimum or maximum scores. We summarized the instruments' quantitative acceptability measures using means and standard deviations. We further described the acceptability as duration of each test, given skewed distributions, as median and interquartile range (IQR).

Internal consistency was evaluated using Cronbach's alpha. We assessed the convergent and discriminant validity of all instruments by evaluating correlations of the test scores with factors hypothesized to be associated with neurocognitive functioning in children at different ages. As such, we tested whether the following factors correlated (convergent validity) or did not correlate (discriminant validity) with neurocognitive functioning as expected from the literature: child age, child height-for-age z-scores (HAZ, normalized numerical assessment of height to age-specific WHO reference values [40], caregiver education, household wealth (derived as the first principal component of the site-specific household assets), proportion of child stimulations (MICS) conducted, and respondent perceived school performance. We quantified these relationships using Pearson's or Spearman's correlation coefficients, the latter when variables were ordinal in nature or there was a visual indication of a violation of the linearity assumption. Linear regression with age adjustment was used to explore associations of scores with non-continuous variables. Finally, age gradients of each instrument's standardized test scores (z-scores) were described using age kernel-weighted local polynomial regression analysis, allowing exploration of non-linear relationships.

Analyses were done using STATA 16 SE (StataCorp, College Station, TX). P-values of <0.05 were considered statistically significant and 95% confidence intervals were reported.

## Ethical approval

The study protocol has been reviewed and approved by Institutional Review Boards at University of California, San Francisco (Reference#: 342371), and Ifakara Health Institute, Tanzania (Reference#: IHI/IRB/No:08–2023), and by the Tanzania National Health Research Ethics Review Committee (Reference#: NIMR/HQ/R.8a/Vol.IX/4204). The study was approved by the President's Office, Regional Administration and Local Government Tanzania (PO-RALG). Additional information regarding the ethical, cultural, and scientific considerations specific to inclusivity in global research is included in the S1 Checklist.

## Results

Of 311 randomly selected children, 261 were included in the study (S1 Fig in S1 Appendix). Children enrolled did not significantly differ from children not enrolled by age. Adult respondents were mostly mothers (70.5%), but there were also other caregivers involved such as grandmothers (10.0%), fathers (8.1%), or aunts (5.0%). Mothers were the primary caregiver in two thirds (65.1%) of all cases. Slightly less than half of children in our analytical sample were female (46.4%), and all age groups from 6 months to 12 years were represented (Table 1). Due to more accurate reporting of children's age during the study compared to the household listing (from which children were randomly selected based on

**Table 1. Demographic characteristics of study sample.**

|  | N | % | Mean | SD |
|---|---|---|---|---|
| **Total** | 261 | 100.0 | | |
| **Child Female** | 121 | 46.4 | | |
| **Age [years]** | | | | |
| <1 | 10 | 3.8 | | |
| 1 to 2 | 27 | 10.3 | | |
| 2 to 3 | 28 | 10.7 | | |
| 3 to 4 | 15 | 5.7 | | |
| 4 to 5 | 29 | 11.1 | | |
| 5 to 6 | 19 | 7.3 | | |
| 6 to 7 | 15 | 5.7 | | |
| 7 to 8 | 21 | 8.0 | | |
| 8 to 9 | 17 | 6.5 | | |
| 9 to 10 | 26 | 10.0 | | |
| 10 to 11 | 21 | 8.0 | | |
| 11 to 12 | 23 | 8.8 | | |
| 12 to 13 | 10 | 3.8 | | |
| **Child attending Kindergarten, Pre-school or Primary School** | | | | |
| No | 90 | 34.5 | | |
| Yes | 171 | 65.5 | | |
| **Highest Caregiver Education** | | | | |
| Less than Primary | 48 | 18.4 | | |
| Primary | 172 | 65.9 | | |
| Secondary | 36 | 13.8 | | |
| Tertiary | 5 | 1.9 | | |
| **Wealth Quintiles** | | | | |
| 1 | 66 | 25.3 | | |
| 2 | 43 | 16.5 | | |
| 3 | 50 | 19.2 | | |
| 4 | 50 | 19.2 | | |
| 5 | 52 | 19.9 | | |
| **Height-for-age*** | | | -1.13 | 1.37 |
| **MICS Stimulations** (proportion) | | | 0.51 | 0.37 |
| **School Performance** (range 1–5) | | | 3.44 | 0.82 |

* For N = 255 out of 261; 6 exceeded plausible WHO chart code range, i.e. HAZ <-6 or >6

age), age year groups differed from the planned 20 children per group. The smallest groups were children aged 12–13 years and below 1 year. About two thirds of children attended kindergarten, pre-school or primary school. The majority of primary caregivers completed primary school (65.9%), and fewer than 2% attained higher education. The mean height-for-age Z-Score (HAZ) of all children was -1.13 (SD: 1.37). Growth faltering was consistent across age strata (S2 and S3 Figs in S1 Appendix). On average, around half (Proportion of 0–6 stimulations practiced: mean, SD: 0.51, 0.37) of the possible child stimulations were practiced in sample households. Caregivers rated their child's school performance between "same" or "somewhat better" than other pupils at the same school (range 1 = much worse to 5 = much better: mean, SD: 3.44, 0.82).

## Feasibility and acceptability

The number of children assessed per instrument was proportional to the instruments' age ranges, and was largest for the EACAB (N = 133 covering school aged children, 6 years and older) followed by the sustained attention tests (PT N = 119, CT N = 118), GSED (N = 79, ages < 4 years), and IDELA (N = 63, ages 4 to 5 years inclusive) (Table 2). Overall, the rates of children refusing and not completing an assessment was low with the highest ones for Sustained Attention CT (5.9%), EACAB Executive Function tests (5.3 and 4.8% respectively) and IDELA (3.2%). Refusal was strongly related to age, with the vast majority of refusals among children from the lowest age year for each instrument group (age of refusal in years: mean, SD: IDELA: 3.58, 0.53, SACT: 8.67, 1.42, EACAB EFWM1: 6.70, 0.58).

We found limited evidence of ceiling or floor effects. For GSED and IDELA, only 1 (1.3%) test child achieved the maximum scores and none scored 0 points (Table 2). For EACAB minimum or maximum scores were rare (<1.5% per sub-test), with notable exceptions of EACAB Planning (3.8% scored no points) and EACAB EFWM1 (9.5% achieved all points). We found substantial proportions of minimum (both >20%) and maximum scores (13.8 and 4.5% respectively) achieved for SAPT and SACT tests. Similar to refusals, minimum and maximum scores were strongly related to age (age of max achieved in years mean, SD: EACAB EFWM1: 10.87, 1.29, SAPT: 9.25, 1.02, SACT: 10.98, 1.43; age of min achieved mean, SD: SAPT: 7.40, 1.69, SACT: 8.97, 1.54).

The findings on assessment burden and feedback from field teams and caregivers indicated the tested instruments generally had high acceptability. The median overall duration of all neurocognitive assessments was 1 hour (median minutes, IQR: 60.1, 35.4–90.7). Duration correlated positively with age (Spearman's correlation coefficient, P-value: 0.76, <0.001). The EACAB had the longest overall duration (median minutes, IQR: 71.7, 31.5) and ran on average longer than IDELA (39.1, 15.6), GSED (24.0, 21.9), SACT (14.7), or SAPT (5.6, 8.6) (Table 2).

**Table 2. Feasibility characteristics of all tested instruments.**

|  | Age group [years] | Number of children completing assessment | Refused or Incomplete* | | Scoring minimum | | Scoring maximum | | Duration [min] | |
|---|---|---|---|---|---|---|---|---|---|---|
|  |  | N | N | % | N | % | N | % | Median | IQR |
| GSED | 0.5 to 4 | 79 | 0 | 0.0 | 0 | 0.0 | 1 | 1.3 | 24.0 | 21.9 |
| IDELA | 3 to 6 | 63 | 2 | 3.2 | 0 | 0.0 | 0 | 0.0 | 39.1 | 15.6 |
| EACAB |  |  |  |  |  |  |  |  |  |  |
| Learning | 6 to 13 | 133 | 0 | 0.0 | 0 | 0.0 | 0 | 0.0 | 11.3 | 7.0 |
| Planning | 6 to 13 | 133 | 0 | 0.0 | 5 | 3.8 | 0 | 0.0 | 13.7 | 12.7 |
| Simultaneous Processing | 6 to 13 | 133 | 0 | 0.0 | 2 | 1.5 | 0 | 0.0 | 10.3 | 6.0 |
| Sequential Processing | 6 to 13 | 133 | 0 | 0.0 | 1 | 0.8 | 0 | 0.0 | 4.0 | 2.5 |
| EFWM1 | 6 to 13 | 133 | 7 | 5.3 | 1 | 0.8 | 12 | 9.5 | 8.6 | 5.9 |
| EFWM2** | 6 to 13 | 126 | 6 | 4.8 | 0 | 0.0 | 0 | 0.0 | 6.4 | 4.2 |
| SAPT | 5 to 11 | 119 | 3 | 2.5 | 25 | 21.6 | 16 | 13.8 | 5.6 | 4.2 |
| SACT | 7 to 13 | 118 | 7 | 5.9 | 29 | 26.1 | 5 | 4.5 | 14.7 | 8.6 |
| Total |  | 261 |  |  |  |  |  |  |  |  |

* Defined as > 50% refused or not completed, or refused to start depending on each test's modalities.

** Only done for children who did not refuse/skip EACAB: EFWM1 test (i.e. refused N = 7).

GSED = The Global Scales for Early Development; IDELA = The International Development and Early Learning Assessment; EACAB = East Africa Cognitive Assessment Battery; EFWM: Executive Function Working Memory; SAPT: Sustained Attention Pencil Tap; SACT: Sustained Attention Code Transmission Test.

Two (0.8%) caregivers expressed concerns about the overall duration of the assessments. On a scale of 1 to 5, tested children and accompanying caregivers rated the difficulty of tests between easy and moderate on average (mean, SD: tested children: 2.70, 0.69; caregivers: 2.81, 0.86). Assessors reported on average between "no" and "some" disturbance during the assessments (mean, SD: 1.70, 0.87). The main challenges for assessors were "other children interrupting" (18.0%), "generally loud environment" (11.5%), "adults interrupting" (10.7%), "child did not pay attention repeatedly" (8.8%), and "child was shy or afraid of the assessor" (5.8%). Caregivers and assessors did not express concerns about the social desirability or acceptability of assessments in the qualitative assessment.

## Internal consistency and convergent validity

We found high internal consistency for GSED, IDELA and the six sub-tests of the EACAB with Cronbach's alpha coefficients of 0.84 or larger (Table 3). The Sequential Processing and Planning sub-tests of the EACAB showed the lowest alphas with 0.84 and 0.88, respectively, and GSED and IDELA having the highest at 0.97 and 0.98, respectively.

The correlation analysis exploring convergent and discriminant validity with factors potentially related to cognitive functioning (at different ages) revealed that age was consistently strongly and statistically significantly correlated with test scores of all instruments (Table 3). GSED and IDELA scores were significantly correlated with height-for-age, IDELA decreasingly so with age (Pearson's correlation coefficients at age 3: 0.65, age 4: 0.43, age 5: 0.28). GSED scores were further correlated with household wealth and child stimulation practices, while IDELA appeared to not correlate with either. IDELA scores were however related to children being in kindergarten or pre-school vs. not (linear age adjusted increase in IDELA score standard deviations: beta, 95%CI, P-value: 0.64, 0.21–1.06, 0.004). Perceived school performance was correlated with all EACAB sub-tests, with the exception of Simultaneous Processing (Table 3). The remaining neurocognitive function measures generally appeared not to be

**Table 3. Internal consistency and convergent validity of all tested instruments.**

| | GSED | IDELA | EACAB | | | | | | SAPT[#] | SACT[#] |
| --- | --- | --- | --- | --- | --- | --- | --- | --- | --- | --- |
| | | | Learning | Planning | Simultaneous Processing | Sequential Processing | EFWM1 | EFWM2 | | |
| **Internal Consistency** | 0.98 | 0.97 | 0.96 | 0.88 | 0.90 | 0.84 | 0.95 | 0.90 | N/A | N/A |
| **Convergent Validity** | | | | | | | | | | |
| Age | 0.79* | 0.62* | 0.32* | 0.43* | 0.39* | 0.36* | 0.58* | 0.50* | 0.50* | 0.45* |
| Height-for-age[&] | 0.34* | 0.27* | 0.02 | 0.01 | -0.07 | 0.03 | 0.06 | -0.06 | 0.17 | -0.01 |
| Caregiver Education[¶] | 0.07 | -0.08 | 0.09 | 0.18* | 0.16 | 0.06 | 0.03 | 0.05 | 0.05 | -0.04 |
| Wealth Quintiles[¶] | 0.27* | 0.04 | 0.09 | 0.28* | 0.02 | 0.14 | -0.04 | -0.05 | -0.01 | 0.01 |
| MICS Stimulations[¶$] | 0.24* | 0.08 | N/A | N/A | N/A | N/A | N/A | N/A | N/A | N/A |
| School Performance[¶@] | N/A | N/A | 0.21* | 0.19* | 0.14 | 0.20* | 0.31* | 0.23* | 0.08 | 0.14 |

* P-value <0.05

& For N = 255 out of 261; 6 exceeded plausible WHO chart code range, i.e. HAZ <-6 or >6

¶ Spearman's correlation coefficients

$ Only for ages 0–6 years

@ Only for ages > = 6 years

# Item level responses not available for alpha testing

GSED = The Global Scales for Early Development; IDELA = The International Development and Early Learning Assessment; EACAB = East Africa Cognitive Assessment Battery; EFWM: Executive Function Working Memory; SAPT: Sustained Attention Pencil Tap; SACT: Sustained Attention Code Transmission Test; MICS: Multiple Indicator Cluster Survey

correlated significantly with the tested factors, except the EACAB: Planning construct with caregiver education and wealth.

## Score gradients

Both GSED and IDELA scores continually increased with age (Fig 1A and 1B, and age correlations in Table 3). GSED and IDELA were significantly correlated for three year old children that were tested with both instruments (Spearman's correlation coefficients, P-value: 0.89, <0.001). The EACAB general intelligence domain scores all increased with age, with lower variability for the Learning and Simultaneous Processing test scores at higher ages (Fig 2). The executive function scores showed more variability across ages, and we observed a tapering off for EFWM2 starting at the age of 10 years. SAPT and SACT attention test scores also increased with age but we observed a flatting of their curves at lower ages, respectively (Fig 3). The SAPT also started tapering off with ages over 8 years. The two sustained attention tests were significantly correlated for 7–10 year old children that were tested with both instruments (Spearman's correlation coefficients, P-value: 0.33, 0.004).

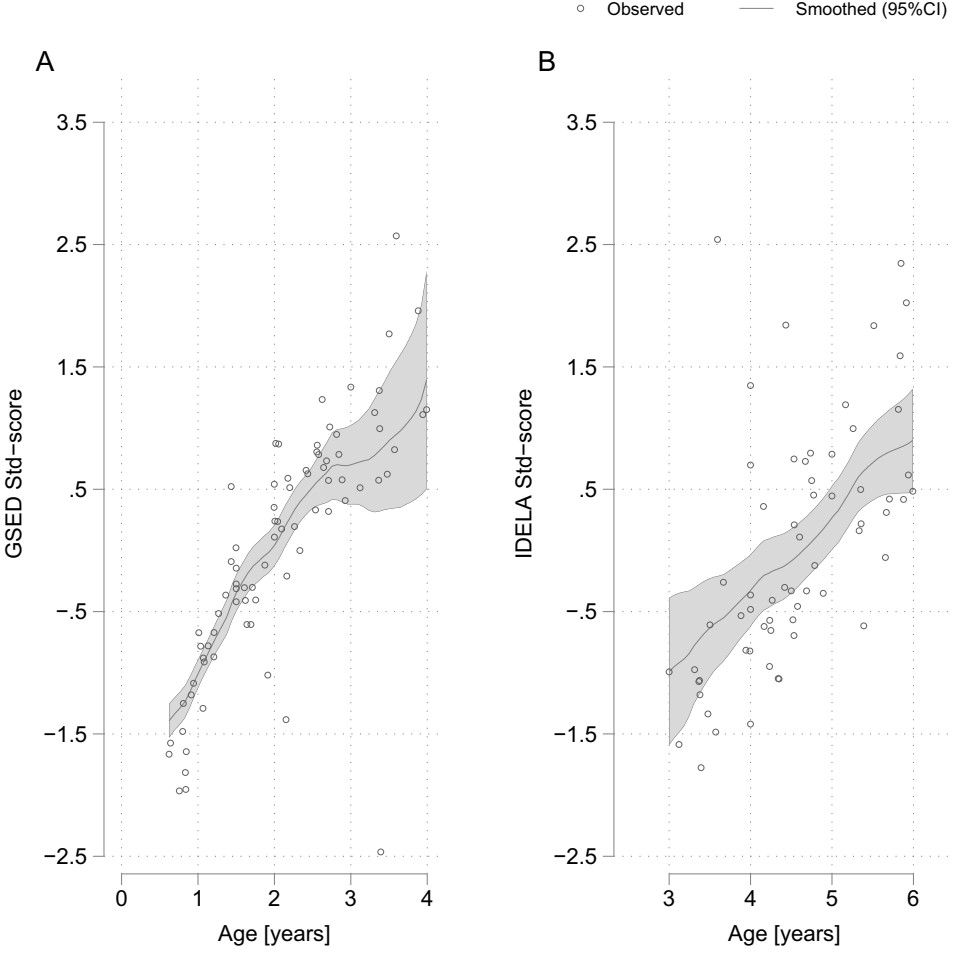

**Fig 1.** A, B. Age gradients for A) GSED (D-score) and B) IDELA (total score) standardized test scores.

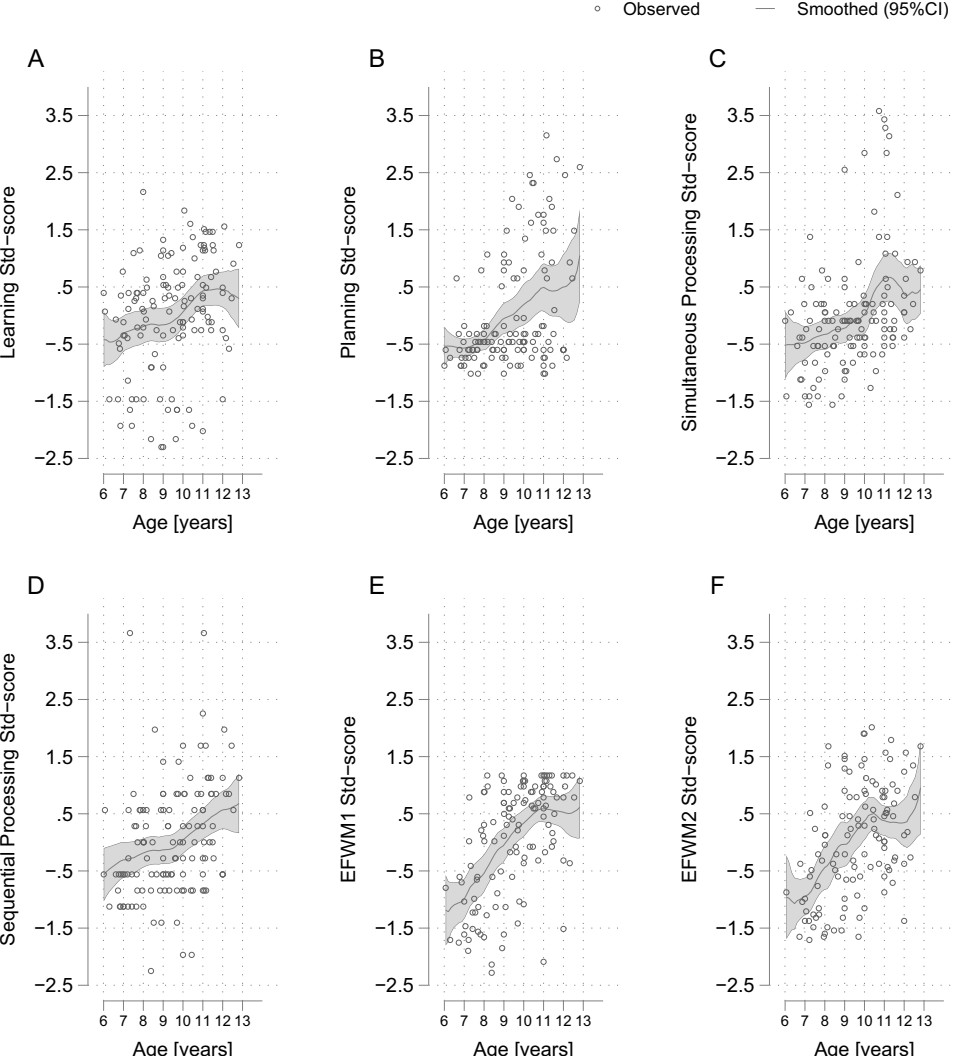

**Fig 2.** A-F. Age gradients for East Africa Cognitive Assessment Battery (EACAB) standardized subtest scores. East Africa Cognitive Assessment Battery standardized subtest scores: A) Learning, B) Planning, C) Simultaneous Processing, D) Sequential Processing, E) Executive Function Working Memory (EFWM) 1, F) EFWM 2.

## Discussion

Our study strongly suggests that measuring neurocognitive functioning of rural Tanzanian children aged 6 months to 12 years of age is feasible and that the currently available tools have the expected psychometric properties. Our tested instruments showed low rates of refusal and extreme scores and adequate test durations, and they were administered with ease via stream-lined data collection processes. The instruments were well-received by participating children and caregivers, highlighting their acceptability in our study population. Moreover, each instrument demonstrated high internal consistency in addition to high convergent and discriminant validity.

The psychometric evaluation and testing of the neurocognitive instruments presented herein sets the stage for their use in evaluating neurocognition in the larger, subsequent CHILD intervention study. Our results support the appropriateness and validity of these

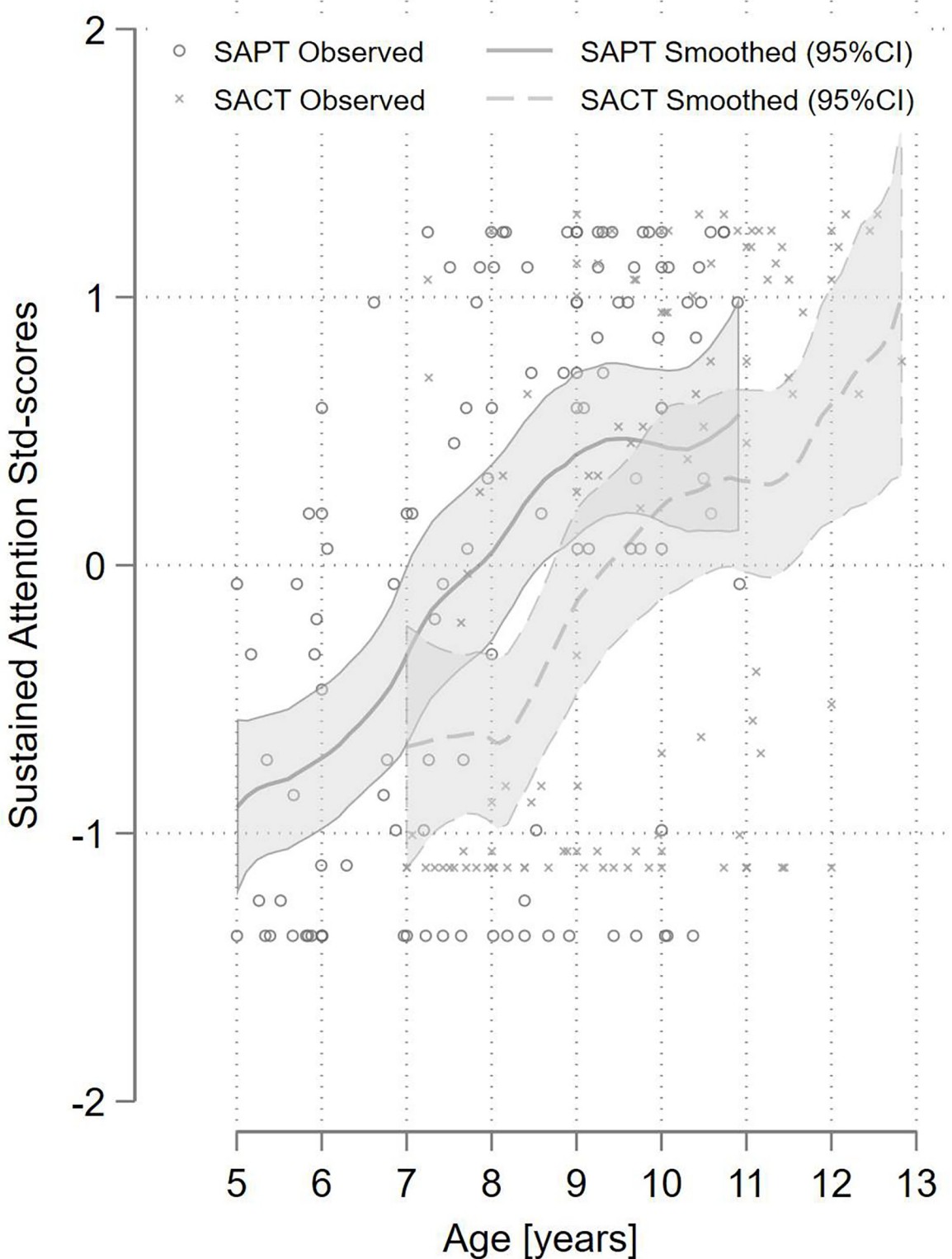

**Fig 3. Age gradients of Sustained Attention: Pencil Tap (SAPT) and Code Transmission (SACT) standardized test scores.**

neurocognitive instruments as tools in elucidating the relationship between LMI and neuro-cognitive functioning in this malaria endemic region. The identification of suitable instru-ments was constrained by guiding factors reflective of the intervention study (e.g., wide age range, local cultural context, resource/time limitations, study site layout, etc.,). However, there have been great strides made in neurocognitive assessments in recent years such that modern instruments which aligned well with these criteria were available at the inception of this study. These advances are due to widespread increased interest in child development and mental health, particularly in underserved populations, over the past decade with growing recognition of the importance of early interventions to promote neurocognitive wellbeing. Given this trend and with support from the current Sustainable Development Goals Agenda [3], there has been rapid development of neurocognitive assessment tools designed for low- and middle-income settings. The selected instruments from this study had therefore already been devel-oped for and validated in Tanzanian or comparable low- and middle-income settings. This allowed for minimal adaptation and adjustment of the tools during pre-testing evaluation. Fur-thermore, all tests were, in comparison to psychometric tests in general, easy to administer and cost efficient, with materials being readily available locally. This finding attests to the thoughtful design and implementation features devised by the creators of these tools.

Since neurocognitive testing will be only part of a wider array of assessments for the CHILD intervention study, an important aspect of our instrument selection was to minimize their time burden. We aimed to keep the average overall duration at around 1 hour but not exceeding 90 minutes per individual child. For children at ages using GSED (<4 years) and IDELA (4–5 years inclusive) this was feasible using the original designed and validated instru-ments. Administering these tools as intended by creators was described in the respective mate-rials to run between 25 to 35 minutes on average. For 5-year-old children, we tested both IDELA and the SAPT, which remained within the time limit. Time constraints became more challenging with increasing age (i.e., tests are longer in general, children progressed farther and did not trigger stop rules, and the SACT test for older kids ran significantly longer than SAPT for younger ones). This necessitated reducing the number of EACAB neurocognitive sub-tests without compromising assessing general intelligence and executive function constructs.

Staff training was intensive given the breadth of materials and exercises needed to correctly administer each age-specific set of instruments. To optimize training sessions, we consolidated shared testing modalities and administration techniques across instruments. Training time was further reduced by using digital mobile data collection applications, relieving the assessors of delays in paper administration and the burden of manual scoring.

The key characteristics of our study sample fell within expected ranges. Distributions of socio-economic factors and child stimulations were plausible for the local region, compared to previous reports from Tanzania [35, 41–43]. The mean HAZ in our analytical sample was approximately 1.13 standard deviations below the expected height for their reference age group, suggesting a level of stunted growth within the population. This is in line with reports on regional stunting rates [44, 45].

Acceptance rates were high for all tested instruments. Few participants exhibited refusal behaviors. The order of testing was the same for each child and we could not explore further whether non-completion rates were related to the test sequence. The incidence of minimum and maximum scores was also low, suggesting minimal floor and ceiling effects. Together with score gradients across age and demographic correlates, this indicates a balanced distribution of test responses across the measurement spectrum. As expected, we observed larger propor-tions of minimum scores for the sustained attention tests. This is partially so because we applied these tools at younger ages than intended by creators to determine ideal cut-offs for

our study. We confirmed that these extreme scores were strongly related to age. The sample fell within the expected and desired test duration ranges, indicative of appropriate test lengths for the target population. Some children were tested with multiple instruments for age-calibrations, naturally exceeding the desired assessment time of 1 hour. The additional burden on these participants allowed for refinement of instruments in the CHILD intervention study and potentially reduce refusal rates, as an assessment for a given child should stay within the desired duration ranges.

Score gradient results suggested all tests captured the age relationship with neurocognitive functioning constructs well (i.e., older ages were clearly related to higher scores as expected). The flattening curves of executive functioning scores (EFWM and sustained attention tests) indicated that tests became either too easy or too difficult at more extreme ages, as more children started scoring near the maximum or minimum scores (and refusal rates increased), respectively. The information from sustained attention tests intersecting age year groups will allow us to estimate optimal age ranges for these assessments in the subsequent intervention study.

Cronbach's alpha coefficients of at least 0.84 for all tests indicate strong reliability and consistency in measuring the underlying constructs [46]. These results substantiate the reliability of selected instruments and support their suitability for assessing the targeted neurocognitive constructs in the rural Tanzanian study population. Due to the data collection process, individual item responses for the two sustained attention instruments were not available for testing their internal consistency.

Our convergent and discriminant validity approach was used to assess the ability of the test to distinguish between the construct being measured and hypothesized related and unrelated variables, respectively. In agreement with our findings for the youngest age group (<4 years, tested with GSED here), socio-economic status, household child stimulation practices and height-for-age were reported to be associated with neurocognitive assessment scores at these ages in prior studies [10, 35]. We found caregiver education was not correlated with GSED scores, similar to findings in GSED validation studies in Tanzania [35]. Our validity findings lend credibility to GSED's usability in our local context. Similarly, IDELA was related to height-for-age, but this was strongly inversely related to age indicating that HAZ becomes less relevant after pre-school ages. Indeed, we did not find correlations of HAZ with any tests scores of school-aged children. IDELA also did not appear to be related to child stimulation as measured by MICS index, but we found children in kindergarten or pre-school scored significantly higher than those who weren't (adjusted for age). This suggested that, in our context, specific learning opportunities might be important at different ages. Also, a previous study showed that associations between stimulation opportunities and cognitive development were not consistent across domains and countries in pre-school children [41], which might explain why we did not detect this using our more comprehensive IDELA construct. Socio-economic factors, like parental education or wealth, have been used as proxies for child neurocognitive development in children, also in low- and middle-income countries [10, 35, 41, 47–49]. At closer examination these previous findings were, however, inconsistent across specific socio-economic factors and their definitions, neurocognitive outcomes, child ages, as well as cultural contexts. We found wealth to be related to neuro-cognition in the youngest children (<4 years of age) but not consistently with older children. This indicated that the constructs our instruments measured might be distinct from external socio-economic influences in school-aged children in our local context. Indeed, consistent significant correlations of EACAB scores and caregiver perceived school performance suggested that unmeasured school-related factors or common causes for better school performance and higher test scores may be more relevant in our specific rural Tanzanian context. This may explain how the findings in our context did not

reveal a positive correlation between socio-economic status and neurocognitive functioning. Alternatively, it is possible that the influence of SES on neurodevelopmental outcomes manifests differently at different ages, which is suggested by our findings. Additionally, the quantification of SES using common household assets may not have adequately captured the full range of local SES variability necessary to detect an association with neurodevelopmental outcomes. Our instruments for neurodevelopmental assessments may not have been sensitive or specific enough to detect differences across socio-economic groups, and we were not equipped to explore non-linear relationships or threshold effects. Other factors, which we did not assess, like environmental factors, genetic predispositions, or access to early childhood education and healthcare services, may have confounded the relationship between SES and neurodevelopmental outcomes. Lastly, our small sample size may limit the ability of our study to reveal this relationship.

While this study provides valuable insights into the usability and validity of our tested instruments, it is important to acknowledge several limitations, in addition to the ones stated in relevant sections above. Although we did not conduct a sample size calculation, the sample size was deemed sufficient to achieve the objectives of the study and to inform the planning of the subsequent larger-scale CHILD intervention study. As our present validation study was explorative in nature and focused on refining study procedures and identifying potential issues with the neurocognitive instruments, a smaller sample size was considered appropriate to manage limited study resources efficiently. As such, given the preliminary nature of this study, our sample size allowed for a more manageable workload in terms of pre-testing, data collection, analysis, and interpretation, thereby facilitating a comprehensive understanding of the intervention's feasibility (regarding neurocognitive assessments) as well as an exploration of psychometric properties of the measurement instruments in the local context. Further, all instruments had been validated and used in Tanzania before the start of this pilot. We aimed to confirm score gradients and effect sizes of internal and convergent validity were consistent with previous reports. Due to more accurate reporting of child ages during the study compared to the preceding household listing (from which children were randomly selected based on age), age year groups differed from the planned 20 children per group. Despite unevenly sized age groups, ages were sufficiently represented, and random selection should ensure that no bias was introduced. This study may be subject to further potential biases associated with sampling and data collection through interviews, which are sometimes dependent on retrospective recall. We attempted to minimize biases using randomized sampling and thoughtful question design. We cannot fully exclude, however, the presence of residual social desirability, response and recall biases, as well as interviewer effects. During the study, we aimed at a more objective assessment of school performance by collecting school report cards, but few caregivers could produce a school report card, so these data were insufficient for use. Additionally, the cross-sectional design of this study limited our ability to infer causality or examine temporal relationships.

Since the neurocognitive instruments used in this study are relatively new, there are some limitations regarding their previous validation. The validation of GSED in Tanzania has been conducted and published in a technical report [35] along with release of v1.0 materials, but peer-reviewed publication is still pending. IDELA has been validated in multiple low- and middle-income countries but not in Tanzania specifically [8, 50, 51]. We used Cronbach's alpha to describe internal reliability of the tested instruments, which tends to under-estimating the true reliability. Guttman's reliability for our hierarchical tests with increasing difficulty could have been employed but would have potentially over-estimated reliability given our relatively small sample size and large number of items.

Despite the limitations, our random and population proportional to size sampling process helped ensure high internal validity, with our study population showing good representativeness of the cultural diversity and socio-economic variability of the broader ward population. Furthermore, we anticipate a high generalizability of these findings to proximal geographic locations, specifically adjacent wards recruiting participants for the CHILD intervention study. The specific context of rural Tanzania limits, however, the generalizability of these findings to other settings.

## Conclusion

The present study indicates that a systematic assessment of general intelligence, executive functioning and more specifically sustained attention constructs in Tanzanian children aged 6 months to 12 years of age is acceptable and feasible with high rates of internal consistency.

## Supporting information

**S1 Checklist. Inclusivity in global research checklist.**
(DOCX)

**S1 Appendix.**
(DOCX)

**S1 Data.**
(ZIP)

## Author Contributions

**Conceptualization:** Susanne P. Martin-Herz, Ally Olotu, Michelle S. Hsiang, Günther Fink.

**Data curation:** Georg Loss, Thabit Athuman.

**Formal analysis:** Georg Loss, Günther Fink.

**Funding acquisition:** Ally Olotu, Michelle S. Hsiang, Günther Fink.

**Investigation:** Georg Loss, Hannah Cummins, Nicolaus Gutapaka, Jane Nyandele.

**Methodology:** Georg Loss, Hannah Cummins, Jane Nyandele, Günther Fink.

**Project administration:** Sylvia Jebiwott, Deborah Sumari, Omary Juma.

**Resources:** Georg Loss, Hannah Cummins, Nicolaus Gutapaka, Jane Nyandele, Deborah Sumari, Omary Juma.

**Software:** Thabit Athuman.

**Supervision:** Michelle S. Hsiang, Günther Fink.

**Writing – original draft:** Georg Loss, Günther Fink.

**Writing – review & editing:** Nicolaus Gutapaka, Sylvia Jebiwott, Deborah Sumari, Thabit Athuman, Omary Juma, Susanne P. Martin-Herz, Ally Olotu, Michelle S. Hsiang.

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
