## [Decision Letter · Decision Letter 0]

29 Oct 2024

PONE-D-24-26872Usability and psychometric properties of a battery of tools to assess intelligence, executive functioning, and sustained attention in Tanzanian childrenPLOS ONE

Dear Dr. Loss,

Thank you for submitting your manuscript to PLOS ONE. After careful consideration, we feel that it has merit but does not fully meet PLOS ONE’s publication criteria as it currently stands. Therefore, we invite you to submit a revised version of the manuscript that addresses the question raised by one of the reviewers during the review process.

Kind regards,

Randall Waechter

Academic Editor

PLOS ONE

Journal Requirements:

3. Thank you for stating the following financial disclosure: This work was supported by NIH/NIAID grant # 1U01AI155315-01A1.  

Additional Editor Comments:

Apologies for the long delay in responding to this submission. Finding reviewers for this manuscript was a significant challenge. I am pleased to report that both reviewers have recommended publication of the manuscript in PLOS ONE, pending a response to one question from one of the reviewers. For this reason, I have recommended Minor Revision. It should not take long to respond to the reviewer's query. This is an important area of work that the International Neuropsychological Society has recently recognized. The INS has called for more effort to both adapt measures for use in low-middle-income country settings and to develop culturally-appropriate measures from within LMICs. The manuscript will make an important contribution to this discussion.

Reviewers' comments:

Reviewer's Responses to Questions

**Comments to the Author**

1. Is the manuscript technically sound, and do the data support the conclusions?

Reviewer #1: Yes

Reviewer #2: Yes

2. Has the statistical analysis been performed appropriately and rigorously? 

Reviewer #1: Yes

Reviewer #2: Yes

3. Have the authors made all data underlying the findings in their manuscript fully available?

Reviewer #1: Yes

Reviewer #2: Yes

4. Is the manuscript presented in an intelligible fashion and written in standard English?

Reviewer #1: Yes

Reviewer #2: Yes

5. Review Comments to the Author

Reviewer #1: I thought this was a well-designed feasibility study that was relevant to the larger goal of the researchers.

I did appreciate that the authors undertook this study to determine the appropriateness of non-local neurocognitive measures for the CHILD study and I think their foresight on the matter should be commended.

One question: What were some of the "similarities" you considered relevant between Yombo and Fukayosi/Kiwanga?

Reviewer #2: The manuscript appears to be technically sound. The five selected instruments (along with pre-testing to assess their applicability and adaptations to fit the study’s context), appear suitable for assessing constructs of general intelligence, executive functioning, and sustained attention in Tanzanian children.

6. PLOS authors have the option to publish the peer review history of their article (what does this mean?). If published, this will include your full peer review and any attached files.

Reviewer #1: No

Reviewer #2: No

---

## [Author Response · Author response to Decision Letter 0]

19 Nov 2024

Thank you very much, we appreciate your time and critical appraisal. We are pleased to hear that the reviewers have recommended our work for publication, pending our response to the remaining query.

REVIEWER #1:

I thought this was a well-designed feasibility study that was relevant to the larger goal of the researchers.

I did appreciate that the authors undertook this study to determine the appropriateness of non-local neurocognitive measures for the CHILD study and I think their foresight on the matter should be commended.

AUTHOR RESPONSE: Thank you very much, we appreciate the reviewer’s time and critical appraisal.

One question: What were some of the "similarities" you considered relevant between Yombo and Fukayosi/Kiwanga?

AUTHOR RESPONSE: Thank you for highlighting this. We added shared characteristics of these wards in this updated text passage (changes underlined) at the beginning of the methods section, line 81-86 of the change-tracked main manuscript:

“This is a cross-sectional validation study conducted in Yombo, Tanzania in 2023. We selected the Yombo ward for its proximity and similarities to the CHILD study area, which is conducted in the adjacent Fukayosi and Kiwangwa wards of the same low malaria transmission region. Yombo shares several relevant characteristics with Fukayosi and Kiwangwa, including demographic makeup, rural settings, socio-economic conditions, and malaria transmission rates. Yombo is rural, consisting of 31 Hamlets with an estimated total population of 11,102 based on the most recent 2022 census.”

REVIEWER #2:

The manuscript appears to be technically sound. The five selected instruments (along with pre-testing to assess their applicability and adaptations to fit the study’s context), appear suitable for assessing constructs of general intelligence, executive functioning, and sustained attention in Tanzanian children.

AUTHOR RESPONSE: Thank you very much, we appreciate the reviewer’s time and critical appraisal.

---

## [Editor Report · Decision Letter 1]

21 Nov 2024

Usability and psychometric properties of a battery of tools to assess intelligence, executive functioning, and sustained attention in Tanzanian children

PONE-D-24-26872R1

Dear Dr. Loss,

We’re pleased to inform you that your manuscript has been judged scientifically suitable for publication and will be formally accepted for publication once it meets all outstanding technical requirements.

Kind regards,

Randall Waechter

Academic Editor

PLOS ONE

---

## [Editor Report · Acceptance letter]

12 Dec 2024

PONE-D-24-26872R1 

PLOS ONE

Dear Dr. Loss, 

I'm pleased to inform you that your manuscript has been deemed suitable for publication in PLOS ONE. Congratulations! Your manuscript is now being handed over to our production team.

Kind regards, 

on behalf of

Dr. Randall Waechter 

Academic Editor

PLOS ONE